# Immune complex deposition promotes NK cell accumulation in the kidney

Abigail De la Cruz[1,2], Marco Garcés[1,3], Emiliano Larios[1,3], Iris K. Madera-Salcedo[1], José C. Crispín[1,4], Florencia Rosetti[1]*

1 Department of Immunology and Rheumatology, Instituto Nacional de Ciencias Médicas y Nutrición Salvador Zubirán, Mexico City, Mexico, 2 Plan de Estudios Combinados en Medicina (PECEM), Facultad de Medicina, UNAM, Mexico City, Mexico, 3 Becario de la Dirección General de Calidad y Educación en Salud, Secretaría de Salud, México, 4 Escuela de Medicina y Ciencias de la Salud, Tecnologico de Monterrey, Monterrey, Mexico

* florencia.rosettis@incmnsz.mx

**Data Availability Statement:** All relevant data are within the manuscript and its Supporting Information files.

## Abstract

In systemic lupus erythematosus, immune complexes deposited in the kidney vasculature represent a potent inflammatory trigger with a high potential to progress to glomerulonephritis and organ failure. These immune complexes can be recognized by multiple effector cells via complement and Fcγ receptors. The transcriptome of CD16-bearing NK cells has been documented in kidneys from patients with SLE. In this study, we show that NK cells accumulate in the kidney in response to immune complex deposition and modulate the behavior of local T cells. Depletion of NK cells transiently ameliorated disease, suggesting NK cells may play a role in lupus nephritis and other immune complex-mediated conditions.

## Introduction

Immune complex (IC) deposition represents the main inflammatory trigger of many types of glomerulonephritis. Patients with systemic lupus erythematosus (SLE), a systemic autoimmune disease, produce copious amounts of autoantibodies that deposit in the glomerular vasculature and accumulate as IC within the glomerular basement membrane [1]. In consequence, about 40% of patients with SLE develop lupus nephritis, a condition in which the glomerular and tubule-interstitial compartments of the kidney develop chronic inflammation that causes structural damage and gradual loss of function. Lupus nephritis is an ominous manifestation of SLE, as up to 20% of patients will evolve to end-stage renal disease within 10 years of diagnosis [2, 3].

Within glomeruli, deposited IC initiate inflammation by activating complement. Immunoglobulin and activated complement represent powerful stimuli for neutrophils who engage IC through complement (*i.e.*, CD11b) and IgG receptors (FcγRs). Neutrophil interaction with glomerular ICs is a key pathogenic element, as its inhibition completely avoids the development of kidney injury in several models of nephritis induced by IC deposition [4, 5]. Nevertheless, neutrophil infiltration is not frequently observed in kidney biopsies from patients with lupus nephritis. Even karyorrhexis, an indicator of local neutrophil death, is an infrequent

**Funding:** This work was supported by CONAHCyT through a research grant (A3-S-39996) to FR and a PhD stipend (1045788) to AC. the funding agency, CONAHCyT, had no involvement in the study design, data collection and analysis, decision to publish, or preparation of the manuscript.

**Competing interests:** The authors have no competing interests to declare.

histological finding [6]. Absence of neutrophils has been suggested to be due to their short life, and/or their transient participation during establishment of disease. Transcriptional studies performed in kidneys from patients with lupus nephritis that have undergone sequential biopsies, have shown expression of neutrophil-related transcripts such as *ITGAM*, *ITGB2*, and *CEACAM8*. Their presence correlates with poor response to standard treatment [7]. Single cell RNA analyses have provided us with a more detailed inflammatory landscape of kidneys in lupus nephritis. The *AMP Consortium*, evaluated the single-cell transcriptome of $CD45^+$ cells from kidney biopsies from 24 patients [8]. Neutrophil transcripts were not present, maybe due to the technical difficulties of maintaining neutrophils for their analysis or due to their absence in these biopsies. However, several FcγR-bearing populations were found, in particular inflammatory $CD16^+$ monocytes. Surprisingly, the most abundant cell population observed were NK cells, particularly $CD56^{DIM}CD16^+$ NK cells, that represented 14% of hematopoietic cells in the kidneys. These cells expressed high levels of cytotoxic transcripts such as *PRF1*, *GZMB* and *GNLY*. Although less abundant, $CD56^{BRIGHT}CD16^-$ cells were also present, and expressed genes suggestive of tissue-residence such as *KIT*, *TCF7*, *IL7R* and *RUNX2* [8]. Due to their pro-inflammatory capacities, $CD56^{DIM}CD16^+$ NK cells could play a relevant role during the initiation or amplification of local inflammation in response to deposited ICs. $CD56^{DIM}CD16^+$ NK cells are efficient producers of cytokines and chemokines and possess potent cytolytic capacities [9]. Classical NK cell activation results from the integration of inhibitory (i.e. KIRs and NKG2A) and activating (NKG2D and NCRs) signals. Their activation threshold is greatly influenced by cytokine stimulation [10]. NK cells can also be activated through CD16 and CD11b, a process that has been mostly studied in the context of antibody-dependent cellular cytotoxicity (ADCC), but their binding to deposited ICs could probably trigger cytotoxicity and IFN-γ secretion as observed in ADCC. Since NK cells lack inhibitory FcγRs, their activation through ICs could represent an intense and long-lasting stimulus within the kidney [9].

Circulating $CD56^{DIM}CD16^+$ NK cells have been described to decrease during active nephritis in patients with SLE. This could be caused by their migration towards target organs. Documentation of their presence in the single cell RNA study of renal biopsies supports this concept [8]. NK cell dynamics during nephritis development has been evaluated in several murine models of SLE. In MRL/*lpr* mice, peripheral NK cells decrease after disease initiation, and this correlates with an increase in the number of NK cells within the kidney [11]. However, the consequences of NK cell infiltration or their potential role within the target organs during disease pathogenesis remain unknown and could be relevant in the establishment of local inflammation. To explore this hypothesis, we evaluated the dynamics of NK cell infiltration in a murine model of lupus nephritis triggered by IC deposition. We show that renal NK cell abundance increases in response to IC deposition and that their depletion transiently decreases the magnitude of glomerular inflammation.

## Materials and methods

### 2.1 Patients and controls

Peripheral blood samples were obtained from patients with SLE that fulfilled the 2019 EULAR/ACR classification criteria for SLE [12] and were diagnosed with proliferative (n = 16) or membranous (n = 14) lupus nephritis by renal biopsy [13]. Sex- and age-matched (± 5 years) healthy controls (HCs) with no family history of autoimmune diseases were recruited as controls (n = 14). Samples from patients and their matched HCs were obtained, processed, and analyzed simultaneously on the day of collection (from 04-09-2021 to 03-09-2022). The study was approved by the IRB of the Instituto Nacional de Ciencias Médicas y Nutrición Salvador Zubirán (IRE-2686). All participants signed informed consent forms.

## 2.2 Mice

C57BL/6 mice obtained from Jackson Labs were housed in specific pathogen-free conditions in accordance to the Instituto Nacional de Ciencias Médicas y Nutrición Salvador Zubirán Institutional Animal Care and Use Committee (IRE-2119 and IRE-1931). Mice between 8–12 weeks of age were used. At indicated times, mice were euthanized in a $CO_2$ chamber. Since glomerulonephritis is asymptomatic, the IACUC committee did not consider it necessary to use analgesia. Mice were monitored daily for signs of discomfort.

## 2.3 Data from AMP Consortium

We used the AMP Consortium website (www.immunogenomics.io) to visualize the abundance of NK cells within CD45+ cells from the kidneys evaluated in the original AMP Consortium study [8].

## 2.4 Nephritis induction and evaluation

Anti-glomerular basement membrane (anti-GBM) glomerulonephritis was induced by immunizing mice with sheep IgG in CFA (day -3) and then intravenously (i.v.) injected with 200 μL of sheep nephrotoxic serum (Probetex) (day 0), as previously reported [14]. Mice were euthanized at indicated times for further analysis. For NK cell depletion experiments, mice were intravenously injected with 200 μg of control IgG or anti-NK1.1 (both from BioXCell) on days -1 and 3. Spot urine samples were collected and urine albumin and creatinine were evaluated by ELISA (Bethyl Laboratories) and a chemical assay (Cayman Chemical), respectively. The results are presented as urine albumin to creatinine ratio. Kidneys were fixed in formalin, and paraffin-embedded for analysis. Tissues sections stained for H&E and PAS were blindly evaluated, and a histological score was given based on endocapillary proliferation, leukocyte infiltration and crescents.

## 2.5 Flow cytometry

Heparinized peripheral blood was obtained by venipuncture. Whole blood was stained with fluorescence-labeled antibodies in the presence of Fc-block. Red blood cells were lysed before analysis.

Murine spleens were disaggregated to obtain a cell suspension, while kidneys were disaggregated after enzyme digestion (collagenase IV and DNAse I; Gibco, Sigma). Red cells were lysed and cell suspension stained for flow cytometry.

Samples were acquired in an Agilent Novocyte Quanteon 4025 flow cytometer and analyzed using the Novoexpress software and FlowJo v10. A complete list of reagents and antibodies used are presented in S1 Table.

## 2.6 Statistical analysis

Results are expressed as mean ± SEM unless noted otherwise. Graphpad Prism 10.0 was used for statistical analyses. Individual tests were chosen based on data distribution.

## Results and discussion

A single cell transcriptomic analysis performed by the *AMP consortium*, showed that CD56$^{DIM}$CD16$^+$ NK cells represent a major fraction of hematopoietic cells in kidneys from patients with lupus nephritis (Fig 1A) [8]. Whether these cells correspond to kidney resident NK cells, or to a peripheral NK cell population that migrated into the inflamed tissue is not clear. Further, their role in the establishment of disease, has not been fully explored.

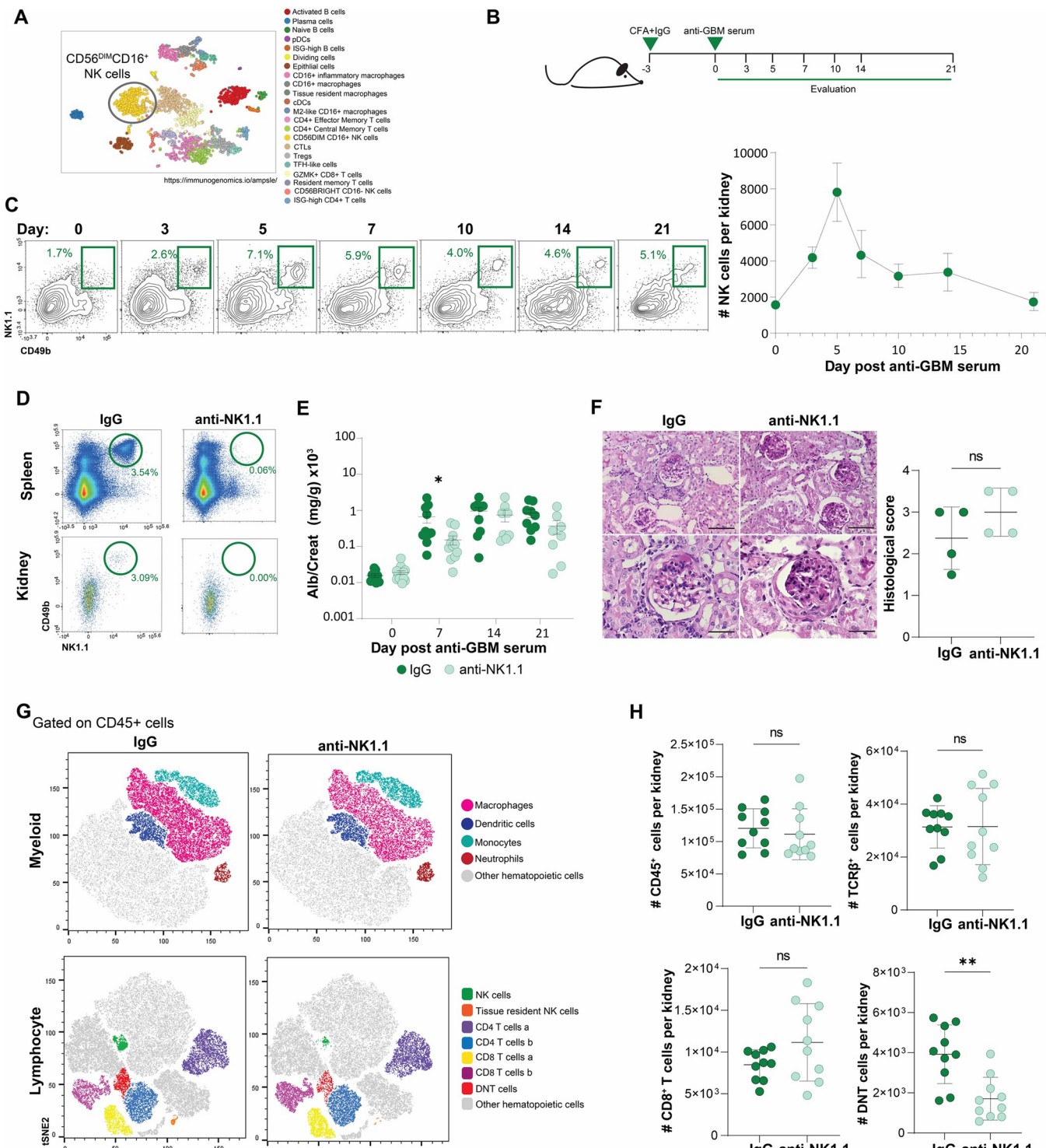

**Fig 1. NK cell abundance increases in kidneys in response to deposited ICs.** (A) Cell clusters identified based on single-cell transcriptional analysis in kidneys from patients with SLE, performed by the AMP Consortium, where NK cells represent a significant population of the hematopoietic cells. Identity of the clusters are specified on the right. (B) Experimental approach of the nephrotoxic nephritis model. (C) Evaluation of NK cells in kidneys upon nephritis induction at the indicated days and absolute NK cell numbers (n = 4–7). (D) Representative plots of NK cells in spleen and kidneys after 7 days of treatment with IgG or anti-NK1.1 antibody. (E) Albuminuria after nephritis induction *p = 0.0305, unpaired t-test. (F) Histological analysis of kidneys of control mice or NK cell-depleted mice; scale bars 80μm (upper panels), and 40 μm (lower panels) (G) Flow cytometry analysis of hematopoietic cell infiltrates in the kidney after nephritis induction in the indicated groups. (H) Total numbers of kidney infiltrating CD45+ cells, T cells, CD8+ T cells and DN T cells in control and NK cell-depleted mice at day 7 of nephritis induction, **p = 0.0011, unpaired t-test.

Because NK cells express FcγRs (i.e. CD16), we decided to evaluate the kinetics of kidney NK cells in an IC-driven glomerulonephritis. For this purpose, mice were subjected to the anti-GBM nephritis model, and kidneys were analyzed at different time points after disease induction (Fig 1B). This disease model allowed us to evaluate the kinetics of NK cells and other leukocytes in a time-controlled manner, as IC deposition initiates immediately after the nephrotoxic serum is injected. Surprisingly, NK cells were rapidly recruited to the kidney (Fig 1C). NK cell numbers peaked at day 5 and gradually decreased (Fig 1C). Most of the kidney NK cells expressed CD11b, an indicator of cellular activation, as CD11b+ NK cells have been shown to be robust producers of IFN-γ, granzyme B, and perforin [15, 16].

To explore the role of NK cell recruitment to the kidney during IC-mediated glomerulonephritis, mice were depleted of NK cells using an anti-NK1.1 antibody 24 hours before anti-GBM serum administration. NK cell depletion occurred within hours and was maintained for at least 21 days (Fig 1D). Interestingly, the NK cell-depleted mice showed a delay in disease development, as reflected by a significant decrease (672.6 ± 229 vs. 152.5 ± 43.4, p = 0.0305) in proteinuria at day 7 (Fig 1E). No significant histological differences were found between groups at day 7 (histological score 1.7 ± 0.2 vs. 1.4 ± 0.24; p = 0.34) nor at day 21 (Fig 1F), suggesting that the effect of NK cell depletion was not sufficiently robust to impact pathology. We evaluated whether kidney cell infiltration was modified by NK cell-depletion. While no changes in total CD45+ cell numbers were observed, a tendency towards an increase in CD8+ T cell infiltration was observed in the NK cell-depleted group. This was associated with a significant decrease in TCRβ+CD4-CD8- T cells (double negative T cells; DNT) (Fig 1G, 1H and S1 Fig). DNT cells are a poorly understood T cell subset, expanded in kidneys of patients with SLE, proposed to correspond to self-reactive pro-inflammatory T cells that produce IL-17, IFN-γ, and TNF-α [17].

CD11b expression was evaluated in the few residual NK cells detected after depletion. Although no differences were found at day 7, a trend to higher expression levels was observed in NK cells from mice that received the anti-NK1.1 antibody (mean fluorescence intensity, MFI: 6544 ± 737 vs. 10884 ± 1864; p = 0.07) at day 21. This observation could be relevant as the initial decrease in proteinuria found in the NK-cell depleted group, could have been partially overcome by an enhanced activation of the few remaining NK cells in the mice in which NK cells were depleted.

In summary, these data suggest that NK cells may migrate into kidneys in response to deposited ICs and may play an initial role in the establishment of renal inflammation.

To determine whether the presence of renal IC affect the numbers or phenotype of circulating NK cells, we recruited patients with lupus nephritis and HCs. Since previous reports have noted an effect of age and sex in circulating NK cells [18, 19], the samples obtained by the HCs were age and sex matched. The demographic and clinical characteristics of the included subjects are presented in S2 Table. Because myeloid cells and NK cells can be activated by their interaction with deposited ICs, we hypothesized that cell proportions or activation markers could differ between these groups of cells. Circulating leukocytes were analyzed as presented in S2 and S3 Figs. No differences in cell proportions or phenotypical markers were observed in the analyzed populations (S4 Fig). However, within the NK cell subset, the proportion of CD56BRIGHTCD16- NK cells, showed a tendency to increase (3.5 ± 0.68 vs. 7.7 ± 2; p = 0.16) while CD56DIMCD16+ population tended to decrease in both groups of patients when compared to healthy controls (58.7 ± 5.5 vs. 53.9 ± 4.2; p = 0.5) (Fig 2A). Accordingly, a negative correlation between SLEDAI at diagnosis with the percentage of circulating CD56DIMCD16+ cells was found (r = -0.446; p = 0.042; Fig 2B). These results are in line with previous reports in which the number of total NK cells is lower in patients with SLE compared with HCs, and this reduction is more accentuated in patients with active disease [15, 20, 21] When specific NK

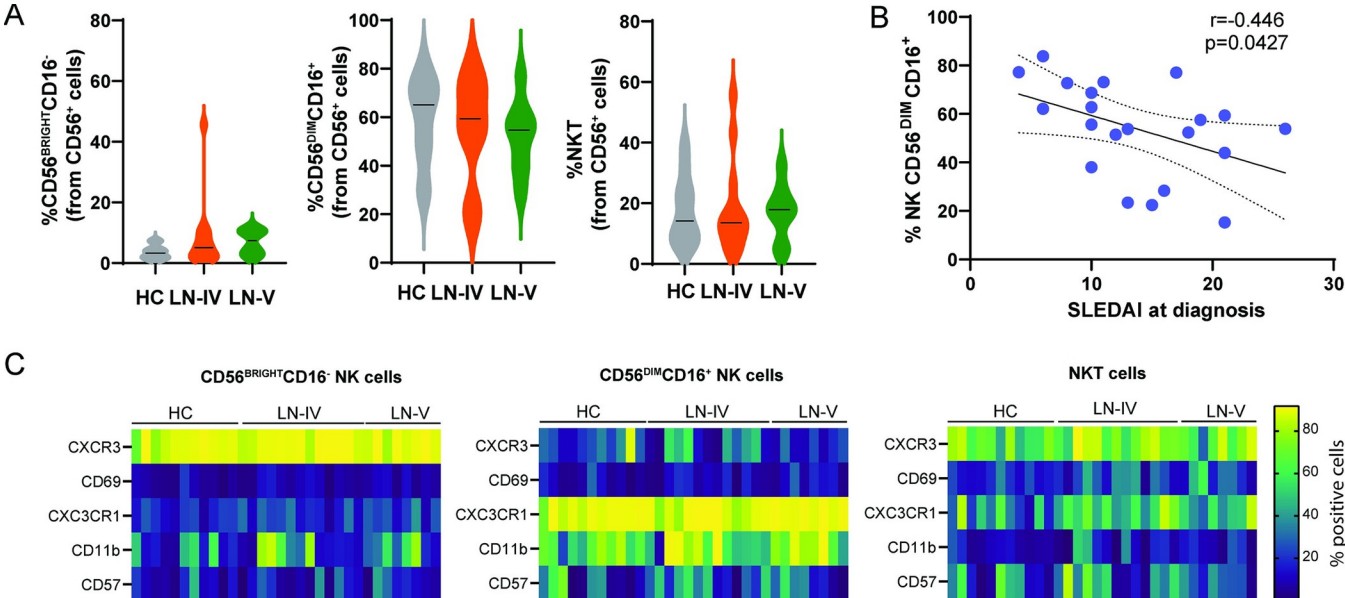

**Fig 2. Evaluation of NK cells in patients with lupus nephritis.** (A) NK cells were evaluated by flow cytometry in peripheral blood of Healthy controls (HC), and patients with type IV (LN-IV) or type V (LN-V) lupus nephritis. (B) Correlation between CD56$^{DIM}$CD16$^+$ cells and SLEDAI at diagnosis (Pearson correlation, p = 0.0427). (C) Heatmaps represent the proportion of cells expressing the indicated activation markers analyzed by flow cytometry. The colorimetric scale represents percentage of NK cell subpopulation expressing the indicated marker.

subsets have been analyzed, a slight decrease in CD56$^{DIM}$CD16$^+$ NK cells [22] and an increase in CD56$^{BRIGHT}$CD16$^-$ NK cells, has also been found, particularly in patients with active disease [23]. In addition to these differences in frequency, NK cells from SLE patients seem to be phenotypically different. Expression of CXCR3 and CD57, molecules involved in migration into the kidney, is decreased in CD56$^{DIM}$CD16$^+$ NK cells from patients with active disease [20, 24–26]. In our analysis, the expression of CXCR3 and CD57 was not modified by the presence of disease (Fig 2C). As previously reported [27], we observed that CD56$^{DIM}$CD16$^+$ NK cells showed a subtle (67.4 ± 5.5 vs. 72.9 ± 5.15) increase in CD11b expression (Fig 2C), suggestive of an active status. CD11b upregulation has been shown to occur in response to increased levels of IL-15 found in active SLE [28]. This could facilitate NK cell migration to target tissues such as the kidney.

Taken together, our data indicate that NK cells are recruited very early into kidneys upon IC deposition, where they contribute to the initiation of local inflammation. The effects of NK cell depletion were very mild in the murine model used in this study. The fact that NK cells are so abundant in kidneys from patients with lupus nephritis suggests a more relevant role than we were able to demonstrate. This inconsistency could be related to certain characteristics of the chosen experimental model. In particular, mice receive a very large load of immune complexes in a very acute manner. This could influence the type of cells that become activated and the kinetics with which they exert their pathogenic effects, masking the role of NK cells. NK cells depletion in the NZBxNZWF1 model of chronic nephritis also delayed the onset of kidney disease, although the authors attributed the effect to the loss of NK T cells [29]. These data, along with the results of the present study suggest that NK cells might contribute to inflammation in the setting of lupus nephritis. However, the importance of NK cells in the human disease must be further evaluated.

## Conclusions

NK cells play an important role in the surveillance against viruses and cancer. However, they can also influence the effector function of other immune cells through the secretion of pro-inflammatory cytokines. In this work, we evaluated circulating leukocytes of patients with lupus nephritis, and found, consistent with previous reports, a decrease in the CD56$^{DIM}$CD16$^{+}$ NK cells, that could be explained by their migration into the inflamed kidney, suggesting their participation in the pathophysiology of lupus nephritis. Using a murine model of acute IC-mediated glomerulonephritis, we show that NK cells infiltrate the kidney during the establishment of disease. By depleting NK cells, we show that these cells can modulate the behavior of T cells within the inflamed tissue and might favor the development of disease. These results highlight the importance of a poorly studied cell population on the pathogenesis of an autoimmune disease.

## Supporting information

**S1 Fig. Lineage markers expression on hematopoietic cells infiltrating kidneys upon induction of nephritis.** Flow cytometry was used to determine leukocyte populations in kidneys at day 7 after induction of the IC-mediated glomerulonephritis. Expression of the indicated (A) myeloid and (B) lymphoid markers on the CD45$^{+}$ cells, is shown as heatmap in T-Distributed Stochastic Neighbor Embedding (tSNE) plots.
(TIF)

**S2 Fig. Gating strategy employed to identify myeloid populations in peripheral blood of the indicated donors.**
(TIF)

**S3 Fig. Gating strategy employed to identify NKT, NK CD56BRIGHTCD16- and CD56DIMCD16+ cells in peripheral blood of the indicated subjects.**
(TIF)

**S4 Fig. Evaluation of frequency and expression of activation markers from neutrophil and monocytes cells in patients with lupus nephritis.** (A) Frequency of Neutrophil and Monocytes was evaluated by flow cytometry in peripheral blood of the indicated groups. (B) Heatmaps represent the proportions of the indicated activation markers analyzed by flow cytometry.
(TIF)

**S1 Table. Antibodies and reagents used in the study.**
(DOCX)

**S2 Table. Demographic characteristics of patients and healthy controls included in the study.**
(DOCX)

**S1 Raw data. Data presented in the figures.**
(XLSX)

## Acknowledgments

We thank MSc. Dámaris P. Romero-Rodríguez from the Laboratorio Nacional Conahcyt de Investigación y Diagnóstico por Inmunocitofluorometría (LANCIDI) of the Instituto Nacional de Enfermedades Respiratorias "Ismael Cosío Villegas" for her help with flow cytometry. We

thank Mariela Contreras, Marysol González-Yañez, Berenice Díaz and Anahí L. Aguilar-López for their help handling the mouse colonies.

## Author Contributions

**Conceptualization:** Abigail De la Cruz, Iris K. Madera-Salcedo, José C. Crispín, Florencia Rosetti.

**Formal analysis:** Abigail De la Cruz, José C. Crispín, Florencia Rosetti.

**Funding acquisition:** Florencia Rosetti.

**Investigation:** Abigail De la Cruz, Marco Garcés, Emiliano Larios, Iris K. Madera-Salcedo, José C. Crispín, Florencia Rosetti.

**Methodology:** Abigail De la Cruz, Marco Garcés, Emiliano Larios, Iris K. Madera-Salcedo, Florencia Rosetti.

**Project administration:** Florencia Rosetti.

**Visualization:** Abigail De la Cruz.

**Writing – original draft:** Abigail De la Cruz, Florencia Rosetti.

**Writing – review & editing:** Abigail De la Cruz, Marco Garcés, Emiliano Larios, Iris K. Madera-Salcedo, José C. Crispín, Florencia Rosetti.

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
