## [Decision Letter · Decision Letter 0]

13 Aug 2024

PONE-D-24-23278Immune complex deposition promotes NK cell accumulation in the kidneyPLOS ONE

Dear Dr. Rosetti,

Thank you for submitting your manuscript to PLOS ONE. After careful consideration, we feel that it has merit but does not fully meet PLOS ONE’s publication criteria as it currently stands. Therefore, we invite you to submit a revised version of the manuscript that addresses the points raised during the review process.

We look forward to receiving your revised manuscript.

Kind regards,

Deborah S. Cunninghame Graham

Academic Editor

PLOS ONE

PONE-D-24-23278

 [This work was supported by CONAHCyT through a research grant (A3-S-39996) to FR and a PhD stipend (1045788) to AC].  

Please include this amended Role of Funder statement in your cover letter; we will change the online submission form on your behalf."

Additional Editor Comments:

Immune complex deposition promotes NK cell accumulation in the kidney by Abigail de la Cruz et. al

This manuscript provides a clear description the contribution of the build up of NK cells build in the kidney to lupus nephritis. I recommend minor revision.

A couple of points from the reviewers bear special emphasis:

Reviewer 1 makes a good point regarding figure 1 "The authors should clarify further (maybe in the methods) what their contribution to figure 1 was. Did they access data and (re-)analyse?"

I look forward to receiving your revised manuscript.

I agree with reviewer 2, that the quality of the figures needs improving to meet publication quality

Reviewers' comments:

Reviewer's Responses to Questions

**Comments to the Author**

1. Is the manuscript technically sound, and do the data support the conclusions?

Reviewer #1: Yes

Reviewer #2: Yes

2. Has the statistical analysis been performed appropriately and rigorously? 

Reviewer #1: Yes

Reviewer #2: Yes

3. Have the authors made all data underlying the findings in their manuscript fully available?

Reviewer #1: Yes

Reviewer #2: Yes

4. Is the manuscript presented in an intelligible fashion and written in standard English?

Reviewer #1: Yes

Reviewer #2: Yes

5. Review Comments to the Author

Reviewer #1: Minor points:

1. Do you have histological analysis of renal tissue from mouse at Day 7 when the mitigating effect of NK depletion is observed, rather than at day 21 when the effect on proteinuria has disappeared?

2. What is the phenotype of the residual NK cells seen in renal material after NK depletion (Fig 1G)? Are these NK cells CD56 bright or dim, are they expressing activation markers?

3. Can you confirm that the correlation of SLEDAI with CD56dim NK cells is with SLEDAI scored at diagnosis rather than at the time of when the sample (for flow cytometry) was taken or are these the same time point?

4. The resolution of the Figures is not great.

Reviewer #2: The authors investigate the present of NK and their subsets in renal tissue from animals with anti-GBM GN as a model for LN. They furthermore investigate peripheral NK cells from SLE patients and controls. They conclude that NK cells infiltrate the kidneys in response to IC deposition where they amplify inflammation.

This referee wonders whether the authors want to discuss the potential future impact of these findings on the prevention of damage, e.g. through modulation of NK cell responses.

The authors should clarify further (maybe in the methods) what their contribution to figure 1 was. Did they access data and (re-)analyse?

What was the sex distribution among healthy controls? Because the number of CD56bright NK cells within peripheral blood decline with increasing age and are more common stable in women, this is important information (doi: 10.1186/1742-4933-3-10; https://doi.org/10.1016/j.mad.2016.04.001). Were SLE patients more commonly female when compared to controls? If so, the authors may even under-estimate differences between groups?

In Figure 2A the authors claim a tendency towards altered NK cell subsets within patients versus controls. The differences appear minor, and it would be helpful to know the p value to assess the statement. In Figure 2C, can the authors quantify their statement that CD11 differs between groups, and a legend to the heat map would be helpful (colour gradient represents expression levels or p values?).

This referee recommends considering this manuscript as a “short article” after addressing minor points outlined above.

6. PLOS authors have the option to publish the peer review history of their article (what does this mean?). If published, this will include your full peer review and any attached files.

Reviewer #1: No

Reviewer #2: No

---

## [Author Response · Author response to Decision Letter 0]

22 Aug 2024

Responses to all reviewer's comments are included in a point-by-point response

---

## [Decision Letter · Decision Letter 1]

1 Oct 2024

Immune complex deposition promotes NK cell accumulation in the kidney

PONE-D-24-23278R1

Dear Dr. Rosetti,

We’re pleased to inform you that your manuscript has been judged scientifically suitable for publication and will be formally accepted for publication once it meets all outstanding technical requirements.

Kind regards,

Deborah S. Cunninghame Graham

Academic Editor

PLOS ONE

Additional Editor Comments (optional):

Thank you for your revised manuscript. I am pleased to inform you that the reviewers are happy that you have addressed all their questions.

Reviewers' comments:

Reviewer's Responses to Questions

**Comments to the Author**

1. If the authors have adequately addressed your comments raised in a previous round of review and you feel that this manuscript is now acceptable for publication, you may indicate that here to bypass the “Comments to the Author” section, enter your conflict of interest statement in the “Confidential to Editor” section, and submit your "Accept" recommendation.

Reviewer #1: All comments have been addressed

Reviewer #2: All comments have been addressed

2. Is the manuscript technically sound, and do the data support the conclusions?

Reviewer #1: (No Response)

Reviewer #2: Yes

3. Has the statistical analysis been performed appropriately and rigorously? 

Reviewer #1: (No Response)

Reviewer #2: Yes

4. Have the authors made all data underlying the findings in their manuscript fully available?

Reviewer #1: (No Response)

Reviewer #2: Yes

5. Is the manuscript presented in an intelligible fashion and written in standard English?

Reviewer #1: (No Response)

Reviewer #2: Yes

6. Review Comments to the Author

Reviewer #1: (No Response)

Reviewer #2: All of my questions have been addressed, and I feel this manuscript is worth publishing. The project is sound, well executed and approrpiately discussed.

7. PLOS authors have the option to publish the peer review history of their article (what does this mean?). If published, this will include your full peer review and any attached files.

Reviewer #1: No

Reviewer #2: **Yes: **Christian Hedrich

---

## [Editor Report · Acceptance letter]

8 Oct 2024

PONE-D-24-23278R1 

PLOS ONE

Dear Dr. Rosetti, 

I'm pleased to inform you that your manuscript has been deemed suitable for publication in PLOS ONE. Congratulations! Your manuscript is now being handed over to our production team.

Kind regards, 

on behalf of

Dr. Deborah S. Cunninghame Graham 

Academic Editor

PLOS ONE